# Oxamate Attenuates Glycolysis and ER Stress in Silicotic Mice

**DOI:** 10.3390/ijms23063013

**Published:** 2022-03-10

**Authors:** Na Mao, Yuhang Fan, Wenjing Liu, Honghao Yang, Yi Yang, Yaqian Li, Fuyu Jin, Tian Li, Xinyu Yang, Xuemin Gao, Wenchen Cai, Heliang Liu, Hong Xu, Shifeng Li, Fang Yang

**Affiliations:** Hebei Key Laboratory for Organ Fibrosis Research, School of Public Health, North China University of Science and Technology, Tangshan 063210, China; namao1991@163.com (N.M.); fyhfyh0705@163.com (Y.F.); liuwenjing_1997@163.com (W.L.); sir1254705936@163.com (H.Y.); yy17732325977@163.com (Y.Y.); lyqewbar@163.com (Y.L.); fuyujinjfy@163.com (F.J.); tiantian__1997@163.com (T.L.); zwns69618@163.com (X.Y.); gaoxm0623@163.com (X.G.); chenwencai1991@163.com (W.C.); 13933499300@163.com (H.L.); xuhong@ncst.edu.cn (H.X.)

**Keywords:** oxamate, glycolysis, ER stress, macrophages, silicosis

## Abstract

Glycolysis and ER stress have been considered important drivers of pulmonary fibrosis. However, it is not clear whether glycolysis and ER stress are interconnected and if those interconnections regulate the development of pulmonary fibrosis. Our previous studies found that the expression of LDHA, a key enzyme involved in glycolysis, was increased in silica-induced macrophages and silicotic models, and it was closely related to silicosis fibrosis by participating in inflammatory response. However, whether pharmacological inhibition of LDHA is beneficial to the amelioration of silicosis fibrosis remains unclear. In this study, we investigated the effects of oxamate, a potent inhibitor of LDHA, on the regulation of glycolysis and ER stress in alveolar macrophages and silicotic mice. We found that silica induced the upregulation of glycolysis and the expression of key enzymes directly involved in ER stress in NR8383 macrophages. However, treatment of the macrophages and silicotic mice with oxamate attenuated glycolysis and ER stress by inhibiting LDHA, causing a decrease in the production of lactate. Therefore, oxamate demonstrated an anti-fibrotic role by reducing glycolysis and ER stress in silicotic mice.

## 1. Introduction

Silicosis is an occupational pneumoconiosis caused by the inhalation of free crystalline silicon dioxide (SiO_2_) or silica dust [1]. When macrophages phagocytize silica particles, they must quickly adapt their metabolism to provide sufficient energy to maintain their immunomodulatory functions, including phagocytosis and the production of inflammatory cytokines and chemokines [2]. This metabolic switch from oxidative phosphorylation to glycolytic metabolism provides an energy source for sustaining inflammatory damage [3]. We previously demonstrated that the levels of key glycolytic enzymes, including hexokinase2 (HK2), pyruvate kinase M2 (PKM2), lactate dehydrogenase A (LDHA), and the lactate concentration were enhanced in silica-induced alveolar macrophage and silicotic models, suggesting that the metabolic switch to glycolysis is an important driving force for the development of silicosis fibrosis [4].

Endoplasmic reticulum (ER) stress and the unfolded protein response (UPR) have been linked to lung fibrosis through the regulation of alveolar epithelial cell (AEC) apoptosis, epithelial–mesenchymal transition (EMT), fibroblast proliferation, myofibroblast differentiation, and M2 macrophage polarization [5,6]. Our previous studies also found that ER stress was abnormally activated in silica-induced A549 cells [7] and MLE-12 cells [8]. Other studies have shown that ER stress and the UPR were critical to the function and phenotypic transformation of macrophages [9]. However, whether silica exposure increases glycolysis and ER stress in macrophages remains unclear.

Lactate, the end product of glycolysis, was once considered a metabolic waste product [10], but several studies have shown that lactate has multiple important biological functions [11,12]. One study showed that the pulmonary release of lactate was directly proportional to the severity of lung injury [13]. In addition, it was recently shown that lactate induced the production of reactive oxygen species (ROS) and promoted the expression of UPR genes [14]. Our previous studies also found that silica induced an increase in extracellular lactate levels in macrophages [4]. However, it is not clear whether lactate can regulate glycolysis and ER stress in NR8383 macrophages. In addition to lactate production, we also found that the expression of LDHA was increased in silica-induced macrophages, and small interfering RNA (siRNA)-*Ldha* inhibited the activation of these macrophages, giving them an anti-inflammatory role [4]. However, whether pharmacological inhibition of LDHA plays an anti-fibrotic role by regulating macrophage function remains unclear.

Oxamate, an amide isostere of pyruvate, is a competitive inhibitor of LDHA [15]. Studies have shown that oxamate inhibits the proliferation of nasopharyngeal carcinoma [16], non-small cell lung cancer [17], and gastric cancer cells [18] and decreases their viability. Therefore, it has been suggested that oxamate may be a promising anticancer agent. However, it is not clear whether oxamate has a therapeutic effect on silicosis fibrosis. In this study, we examined the effects of oxamate on the regulation of glycolysis and ER stress in macrophages and a silicotic mouse model and used these data to elucidate the therapeutic mechanism of oxamate in the treatment of silicosis fibrosis.

## 2. Results

### 2.1. Silica Increased ER Stress in Macrophages

In our preliminary study, we found that SiO_2_ induced ER stress in A549 cells [7] and MLE-12 cells [8]. As a follow-up, this study was aimed at further investigating whether ER stress was also manifested in silica-induced macrophages. As shown in Figure 1A, the fluorescence intensity of phospho-PKR-like ER kinase (p-PERK) was significantly enhanced in a dose-dependent manner by supplementation of the macrophages with 100, 200, and 250 µg/mL SiO_2_. Furthermore, exposure of the macrophages to SiO_2_ induced an increase in the expression of phospho-inositol-requiring enzyme 1α (p-IRE-1α) (Figure 1B). We also found that SiO_2_ significantly increased intracellular ROS production (Figure 1C). Western blotting analysis indicated that the expressions of p-PERK, p-IRE-1α, and phospho-eukaryotic initiation factor 2 alpha (p-eIF-2α) were upregulated in a dose-dependent manner by supplementation of the macrophages with 100, 200, and 250 µg/mL SiO_2_ (Figure 1D and Appendix A). These results suggested that silica induced ER stress in macrophages.

### 2.2. Oxamate Attenuated the Enhancement of Glycolysis and ER Stress in Silica-Induced Macrophages

Since glycolytic reprogramming and ER stress are critical for the functional and phenotypic transformation of macrophages, we further investigated whether the targeted inhibition of LDHA had an effect on the regulation of glycolysis and ER stress. Oxamate alleviated the silica-induced increase in the expression of p-PERK and p-IRE-1α as well as the increase in ROS production (Figure 2A–C). Similarly, oxamate treatment reduced the extent of ER stress by inhibiting the silica-induced upregulation of p-PERK and p-IRE-1α, but not by inhibiting p-eIF-2α (Figure 2D and Appendix A). To investigate the effect of oxamate on metabolic changes in macrophages, we assessed the changes in the extracellular acidification rate (ECAR), lactate concentration, and the mitochondrial oxygen consumption rate (OCR) after treatment of the silicotic macrophages with oxamate. After exposure of the macrophages to silica, we detected increases in both the ECAR and lactate concentration but a decrease in the OCR, indicating the transition from oxidative phosphorylation to aerobic glycolysis. However, these effects were reversed upon treatment of the silica-exposed macrophages with oxamate (Figure 3).

### 2.3. Lactate Promoted ER Stress in Macrophages

To determine the effect of lactate on the expression of ER stress-related factors in NR8383 cells, we stimulated NR8383 cells with different concentrations of lactate. As shown in Figure 4A,B, the expression of p-PERK and p-IRE-1α gradually increased with the increase in the lactate concentration. It was previously reported that exogenous lactate stimulation resulted in an increase in intercellular ROS production in skin fibroblasts [19]. Our study corroborated this finding, as lactate was also found to stimulate the production of ROS in NR8383 cells (Figure 4C). Supplementation of exogenous lactate triggered an increase in the expression of p-PERK, p-IRE-1α, and p-eIF-2α in NR8383 cells in a dose-dependent manner (Figure 4D and Appendix A). Taken together, these results indicated that lactate played a key role in the activation of ER stress in macrophages.

### 2.4. Oxamate Attenuated the Lactate-Induced Enhancement of Glycolysis and ER Stress in Macrophages

Next, we used oxamate to inhibit LDH and prevent the conversion of pyruvate to lactate, reducing the production of lactate. We found that treatment with oxamate inhibited the enhanced expression of p-PERK and p-IRE-1α and ROS production induced by exogenous lactate (Figure 5A–C). In addition, treatment of the silica-exposed macrophages with oxamate inhibited the activation of ER stress by targeting p-PERK and p-IRE-1α rather than p-eIF-2α (Figure 5D and Appendix A). Next, we sought to verify whether oxamate regulated lactate-induced metabolic changes. Treatment of the macrophages with oxamate resulted in lower levels of ECAR, a reduced production of lactate, and higher levels of OCR due to the inhibition of LDHA (Figure 6). These findings indicated that oxamate alleviated the lactate-induced increase in glycolytic metabolism and ER stress by inhibiting LDHA.

### 2.5. Oxamate Reduced Glycolysis and ER Stress in Silicotic Mice

Since we observed that oxamate inhibited glycolysis and ameliorated ER stress in macrophages, we considered whether oxamate could mitigate fibrotic responses to silica in vivo. To test this hypothesis, we administered different doses of oxamate to mice exposed to silica. Treatment of the silicotic mice with 100 mg/kg or 500 mg/kg oxamate per day for 4 weeks led to a significant reduction in the expression of LDHA and attenuation of fibrotic remodeling in the silica-exposed lungs of the mice, as assessed by HE. and Sirius Red histological staining of the lung tissue (Figure 7A–C). Consistent with our in vitro observations, treatment of the mice exposed to silica with 100 mg/kg or 500 mg/kg oxamate led to the reduced expression of p-PERK and p-IRE-1α but had no effect on p-eIF-2α; in addition, the effects of the 500 mg/kg oxamate dosage on the expression of the two proteins were more significant than the 100 mg/kg dosage (Figure 7D–F and Appendix A). Taken together, these data supported that oxamate played an anti-fibrosis role in vivo by inhibiting glycolysis and ER stress in macrophages.

## 3. Discussion

Macrophages rely heavily on glycolysis for bioenergetics [20]. Silica is a classical activator of glycolysis in macrophages [21]. After activation, nutrient flux changes rapidly, and glycolysis becomes the main pathway by which metabolic intermediates are biosynthesized in macrophages to support the synthesis of ribose, amino acids, and fatty acids to sustain their immunomodulatory activities [2]. However, some studies have suggested that bone marrow-derived macrophages mainly rely on glycolysis when participating in inflammatory responses, while tissue-resident macrophages mainly rely on oxidative phosphorylation, and glycolysis has been maintained at a low level [22,23]. Our previous study found that tissue-resident macrophages also mainly rely on glycolysis for the production of energy to perform biological functions [4]. We observed that glycolysis contributed to the regulation of macrophages in silicosis by promoting the activation of M1/M2 macrophages.

The activation of ER stress promoted the polarization of macrophages into the pro-inflammatory phenotype, and this polarization was inhibited by the ER stress inhibitor 4-phenylbutyrate (4-PBA) as well as GSK2656157, an inhibitor of PERK [24]. Sustained activation of the UPR signaling pathways and upregulation of transcription factors, such as C/EBP homologous protein (CHOP) and activating transcription factor 4 (ATF4), induced liver injury [25]. Inhibition of the ER stress-associated IRE-1/XBP-1 (ER processing) signaling pathway suppressed M1 polarization and ameliorated lipopolysaccharide (LPS)-induced lung injury [26]. We observed previously that SiO_2_ induced ER stress-associated apoptosis in A549 cells [7]. In addition, supplementation with 4-PBA inhibited the activation of senescence signaling in silica-induced MLE-12 cells [8]. In this study, we discovered that the proteins related to ER stress, p-PERK, p-IRE-1α, and p-eIF-2α, were up-regulated in the silica-exposed macrophages, suggesting that silica activated ER stress in the macrophages.

More and more evidence that the activation of macrophages causes the immune cells to switch their metabolism from oxidative phosphorylation to aerobic glycolysis, resulting in the increased production of lactate, has surfaced in recent years [27,28]. The increased lactate provides energy for the growth of the macrophages. Studies have shown that the functions and polarization of M2 macrophages could be directly regulated by lactate in the tumor microenvironment (TME) via hypoxia inducible factor-1 (HIF-1α)-dependent metabolic reprogramming [29]. Lactate derived from myofibroblasts promoted the polarization of macrophages into the pro-fibrotic phenotype, thereby promoting pulmonary fibrosis [30]. Interestingly, ROS production was increased after exposure of SH-SY5Y cells to exogenous lactate in a dose-dependent manner [14]. Additionally, lactate promoted an increase in the expression of DNAJ chaperones and UPR genes, including glucose-regulated protein 78 (GRP78)/BiP, XBP1, and heat shock-related 70 kDa protein 2 (HSPA2), which are elements well-known to be directly involved in ER stress responses [14]. In this study, we also found that exogenous lactate stimulated the activation of glycolysis, ER stress, and the production of ROS in macrophages, indicating that lactate was an integral metabolic precursor that provided macrophages with energy for regulating the stress response.

Increasing evidence has indicated that glycolysis and the UPR, which are two independently activated signaling pathways, may function cooperatively to maintain the biological functions of macrophages [31]. Studies have shown that the UPR transcription factor ATF4 upregulated glycolytic enzymes and LDH in Drosophila melanogaster S2 cells to produce more lactate [32]. Similarly, when Drosophila melanogaster S2 cells were treated with ER stress activators dithiothreitol (DTT) and tunicamycin (Tm), the expression of genes encoding glycolytic enzymes was upregulated, whereas the expression of genes encoding TCA cycle and mitochondrial respiratory chain enzymes was downregulated, and the expression of LDHA was significantly increased. Another human glioma study also found that the upregulation of UPR targets and the increase in glycolysis was related to poor patient prognosis [33]. Glycolytic intermediates produce important intermediates for protein N-glycosylation in the ER and ATP for enabling protein folding and processing [34,35]. However, other studies have demonstrated opposite views. In one study, the UPR reduced glycolysis and mitochondrial respiration via IRE1 signaling [36]. In addition, GRP78 overexpression suppressed the expression of glycolytic proteins, including LDHA, pyruvate dehydrogenase kinase 1 (PDK1), and c-Myc [37]. In another study, pyruvate down-regulated the expressions of GRP78, CHOP, ATF4, and p-eIF2a and ameliorated ER stress in HK-2 cells [38]. These contradictive results might be caused by the differences in cell type or the severity of ER stress and/or glycolysis in the different studies. Therefore, it is important to further explore the role of glycolysis and ER stress in macrophages to better understand the mechanisms of interaction between these processes.

In previous studies, the levels of LDHA and its metabolic product, lactate, were increased in silica-induced rats and mice, indicating that LDHA might be an effective potential therapeutic target for silicosis [4]. Studies have shown that the LDHA inhibitor gossypol inhibited ionizing radiation and bleomycin-induced pulmonary fibrosis [39,40]. Oxamate, a competitive inhibitor of LDHA, has been demonstrated to have a promising anticancer effect [17]. However, it was not clear whether oxamate had an effect on ameliorating pulmonary fibrosis. In this study, we found that oxamate treatment maintained macrophage homeostasis by ameliorating the increase in glycolysis and ER stress induced by exposure of the macrophages to silica, thereby endowing oxamate with anti-fibrotic properties.

## 4. Materials and Methods

### 4.1. Animal Experiments

All animal protocols were reviewed and approved by the Committee on the Ethics of North China University of Science and Technology (LX2019033) and complied with the U.S. National Institutes of Health Guide for the Care and Use of Laboratory Animals [41].

To explore the anti-fibrotic effect of oxamate, specific pathogen-free male C57BL/6 mice weighing 15 ± 3 g at 8 w of age were purchased from Vital River Laboratory Animal Technology (Beijing, China). All mice were housed in a specific pathogen-free facility and were maintained under conditions of constant temperature (23–25 °C), humidity (40–50%), and a 12 h light/dark cycle. The mice were randomly divided into four groups (*n* = 10 each) as follows: (1) control group, (2) silica group, (3) silica plus oxamate (100 mg/kg) group, and (4) silica plus oxamate (500 mg/kg) group. In these studies, silicosis was induced by intratracheal instillation of 50 μL of a silica suspension (5 mg/mouse, s5631, Sigma-Aldrich, St. Louis, MO, USA). Subsequently, after exposure of the mice to silica for 4 weeks, the mice in the two silica plus oxamate groups were treated with either 100 mg/kg or 500 mg/kg oxamate (01089157, damas-beta, Shanghai Titan Scientific Co., Ltd., Shanghai, China) by intraperitoneal injection daily until 8 weeks, and the lungs were harvested and stored at −80 °C until analysis.

### 4.2. Pulmonary Histopathologic Observation

Lung tissue slices (4 μm thick) were stained with Hematoxylin–Eosin (H–E.) stain (BA4025, Baso Diagnostics Inc., Zhuhai, China) to observe the pathological changes between mice groups and with Sirius red stain (A0002-10, Report Company, Hebei, China) to perform semi-quantitative analysis of the collagen levels in the lung tissue. The area of silicosis nodules and collagen stained by Sirius red were counted by Image-Pro Plus 6.0 software package (Media Cybernetics, Rockville, MD, USA) and were homogenized by the total area of lung section.

### 4.3. Cell Culture and Treatment

NR8383 alveolar macrophages was acquired from the Cell Bank of the Chinese Academy of Sciences (Shanghai, China). The cells were grown as monolayer cultures routinely in Ham’s F-12K medium (L450 KJ; Shanghai BasalMedia Technologies Co., Ltd., Shanghai, China) in an incubator with a 5% CO_2_ atmosphere at 37 °C. The cells were subjected to serum starvation for 24 h and then treated with different doses of SiO_2_ or lactate (L14500; Alfa Aesar, Shanghai, China) in serum-free medium for 24 h. The culture medium was supplemented with 50 mM solution of oxamate sodium 1 h prior to SiO_2_ or lactate treatment, after which the cells were incubated for 24 h. Oxamate sodium was diluted with sterile water.

### 4.4. Immunohistochemistry (IHC) and Immunofluorescence Staining (IF)

Briefly, paraffin sections of the lung tissue and cell slides underwent high-pressure antigen retrieval followed by blocking with 0.3% (*w*/*v*) H_2_O_2_ for 15 min to quench the endogenous peroxidases. The samples were then incubated with anti-p-IRE-1α (S724) (ab48187, Abcam) primary antibodies overnight at 4 °C. The next day, the samples were incubated with a secondary antibody (PV-6000, Beijing Zhongshan Jinqiao Bio-Technology Co., Ltd., Beijing, China) at 37 °C for 30 min. The immunoreactivity of the samples was visualized with 3,3-diaminobenzidine (DAB) (ZLI-9018; Beijing Zhongshan Jinqiao Bio-Technology Co., Ltd.,).

IF staining was performed as previously described [42]. Paraffin sections of the lung tissue and cell slides were incubated with anti-p-PERK (Thr982) (DF7576, Affinity) and anti-LDHA (DF6280, Affinity) antibodies overnight at 4 °C. The samples were then incubated with the secondary antibody at 37 °C for 60 min. The cellular nuclei were stained with DAPI (8961s; Cell Signaling Technology, Inc., Danvers, MA, USA).

### 4.5. Western Blot

Western blot was performed using a previously published protocol [43]. The primary antibodies used were as follows: anti-p-PERK (DF7576, Affinity), anti-PERK (ER64553, HuaBio), anti-p-IRE-1α (S724) (ab48187, Abcam), anti-IRE1α (A00683-1, Boster), anti-p-eIF-2α(S51) (ET1603-14, HuaBio), anti-eIF-2α (ET7111-34, HuaBio), anti-Col I (ab34710, Abcam), anti-LDHA (DF6280, Affinity), and β-actin (AC026, ABclonal). After supplementation with the primary antibodies, the lung tissues and cells samples were incubated with goat anti-rabbit or anti-mouse secondary antibodies (074-1506/074-1806; Kirkegaard & Perry Laboratories, Gaithersburg, MD, USA) at a concentration of 1:5000 for 1 h. The target bands were visualized using the ECL prime Western blotting detection reagent.

### 4.6. Real-Time Cell Metabolism Assay

An XF-96 Extracellular Flux Analyzer (Seahorse Bioscience, Agilent Technologies, Billerica, MA, USA) was used to analyze ECAR, a representation of the glycolytic rate and OCR, a representation of mitochondrial respiratory activity in the NR8383 cells. Baseline measurements were taken before commencement of the assay, after which 10 mM glucose, 1 µM oxidative phosphorylation inhibitor oligomycin, and 50 mM glycolysis inhibitor 2-deoxy-D-glucose (2-DG) were sequentially injected into each well at pre-determined time points to measure the ECAR. For the measurement of the OCR, 1 µM oligomycin, 1 µM protonophore trifluoromethoxy carbonylcyanide phenylhydrazone (FCCP), and 1 µM mitochondrial complex I inhibitor rotenone plus the mitochondrial complex III inhibitor antimycin A (Rotenone/Antimycin A) were sequentially injected into each well. The data were analyzed using the Seahorse XF-96 Wave software.

### 4.7. Extracellular Lactate Measurement

The lactate concentration in NR8383 alveolar macrophage culture medium was determined using a lactate assay kit (Nanjing Jiancheng Bioengineering Institute, Nanjing, China) according to the manufacturer’s instructions.

### 4.8. Measurement of the ROS Levels

The levels of intracellular ROS were measured using DCFH-DA (Cayman Chemical, Ann Arbor, MI, USA). The NR8383 cells were incubated with 10 µmol/L DCFH-DA for 30 min at 37 °C. The Image-Pro Plus 6.0 software package (Media Cybernetics, Rockville, MD, USA) was used to quantify the integral optical density (IOD) of ROS fluorescence based on the images acquired by the fluorescence microscope.

### 4.9. Statistical Analysis

Data analysis was conducted using the SPSS 20.0 statistical software (SPSS Inc., Chicago, IL, USA). The data were expressed as the mean ± standard deviation (SD). Multiple comparisons were performed using one-way analysis of variance (ANOVA) followed by the Tukey test for post hoc analysis. *p* values < 0.05 were considered statistically significant. All calculations were performed using GraphPad Prism 8 software (GraphPad Software, La Jolla, CA, USA).

## Figures and Tables

**Figure 1 ijms-23-03013-f001:**
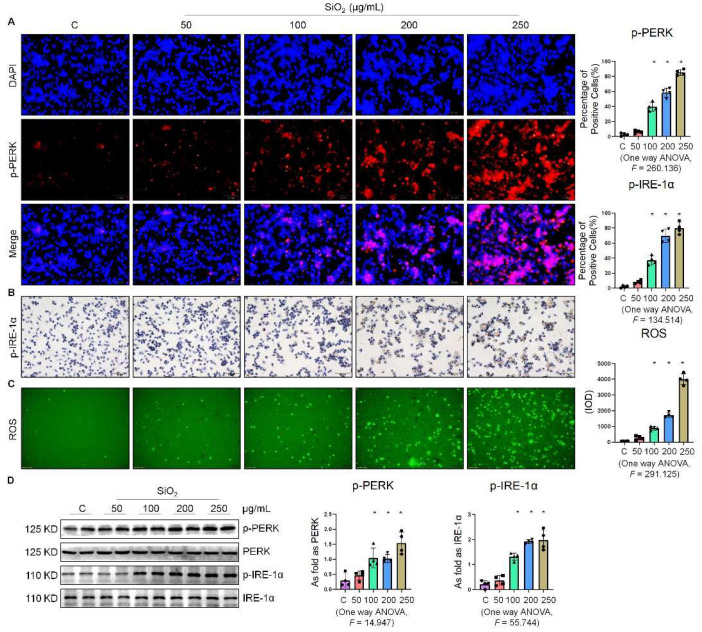
Silica increased ER stress in macrophages. (**A**) Expression of p-PERK in NR8383 cells treated with SiO_2_ at different doses observed by immunofluorescence (IF) staining (scale bar = 50 µm). (**B**) Expression of p-IRE-1α in NR8383 cells treated with SiO_2_ at different doses observed by immunohistochemistry (IHC) staining (scale bar = 50 µm). (**C**) Intracellular ROS production in NR8383 cells treated with SiO_2_ at different doses observed by 2,7-dichlorodihydrofluorescein diacetate (DCFH-DA) staining (scale bar = 100 µm). (**D**) Protein expression of p-PERK and p-IRE-1α in NR8383 cells treated with SiO_2_ at different doses measured by Western blotting. * Compared with control group, *p* < 0.05. Data are presented as the mean ± SD, *n* = 4 per group.

**Figure 2 ijms-23-03013-f002:**
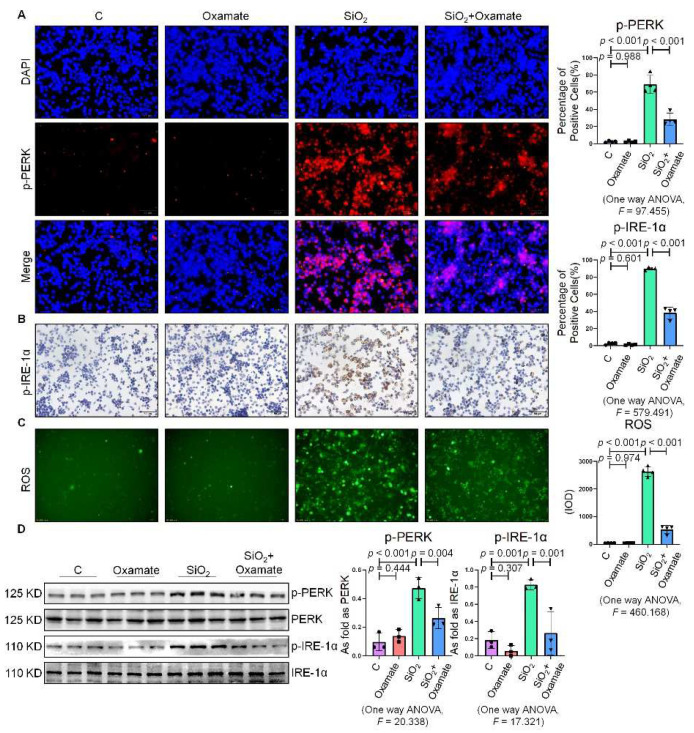
Oxamate attenuated the enhancement of ER stress in silica-induced macrophages. (**A**) Expression of p-PERK in NR8383 cells observed by IF staining (scale bar = 50 µm). (**B**) Expression of p-IRE-1α in NR8383 cells observed by IHC staining (scale bar = 50 µm). (**C**) Intracellular ROS production in NR8383 cells observed by DCFH-DA staining (scale bar = 100 µm). (**D**) Protein expression of p-PERK and p-IRE-1α in NR8383 cells treated with oxamate, SiO_2_, and SiO_2_ plus oxamate measured by Western blotting. Data are presented as the mean ± SD, *n* = 3 per group.

**Figure 3 ijms-23-03013-f003:**
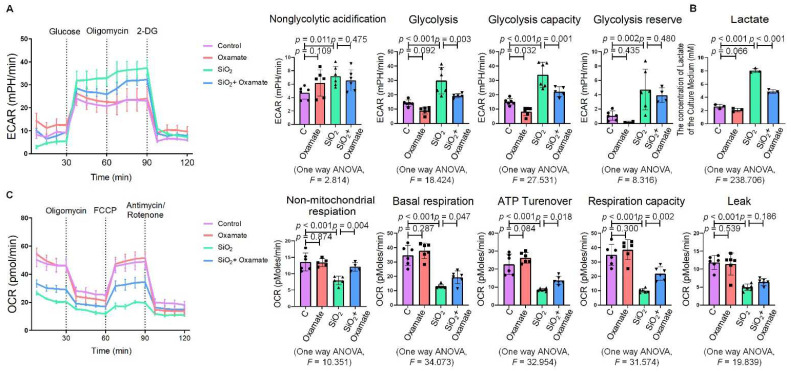
Oxamate attenuated the enhancement of glycolysis in silica-induced macrophages. (**A**) Glycolysis flux was examined by measuring the ECAR using the Seahorse analyzer. Data are presented as the mean ± SD, *n* = 6 per group. (**B**) The lactate concentration in the culture medium was detected using a lactate assay kit. Data are presented as the mean ± SD, *n* = 3 per group. (**C**) The OCR values were measured using the Seahorse analyzer. Data are presented as the mean ± SD, *n* = 6 per group.

**Figure 4 ijms-23-03013-f004:**
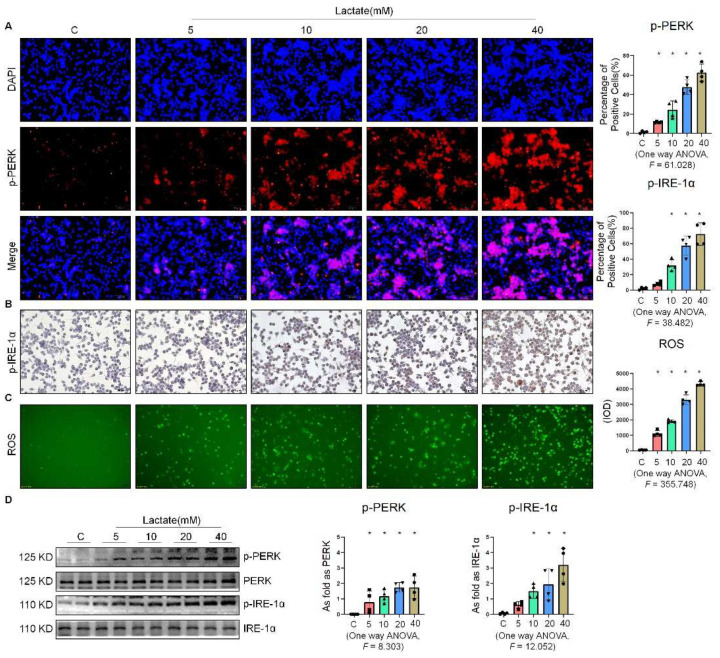
Lactate promoted ER stress in macrophages. (**A**) Expression of p-PERK in NR8383 cells treated with lactate at different doses observed by IF staining (scale bar = 50 µm). (**B**) Expression of p-IRE-1α in NR8383 cells treated with lactate at different doses observed by IHC staining (scale bar = 50 µm). (**C**) Intracellular ROS production in NR8383 cells treated with lactate at different doses observed by DCFH-DA staining (scale bar = 100 µm). (**D**) Levels of p-PERK and p-IRE-1α in NR8383 cells treated with lactate at different doses measured by Western blotting. * Compared with control group, *p* < 0.05. All data are presented as the mean ± SD, *n* = 4 per group.

**Figure 5 ijms-23-03013-f005:**
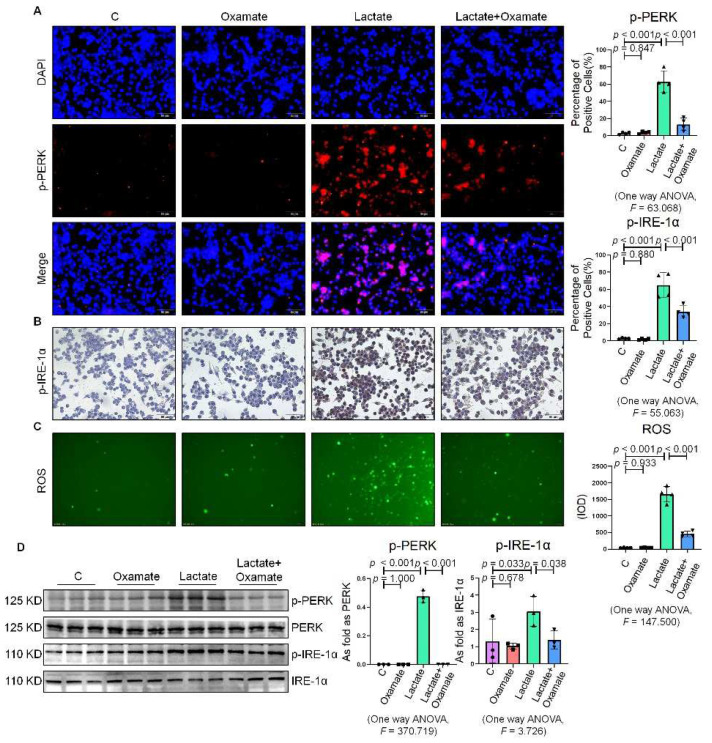
Oxamate attenuated the lactate-induced enhancement of ER stress in macrophages. (**A**) Expression of p-PERK in NR8383 cells observed by IF staining (scale bar = 50 µm). (**B**) Expression of p-IRE-1α in NR8383 cells observed by IHC staining (scale bar = 50 µm). (**C**) Intracellular ROS production in NR8383 cells by using DCFH-DA staining (scale bar = 100 µm). (**D**) Protein expression of p-PERK and p-IRE-1α in NR8383 cells treated with oxamate, lactate, and lactate plus oxamate measured by Western blotting. Data are presented as the mean ± SD, *n* = 3 per group.

**Figure 6 ijms-23-03013-f006:**
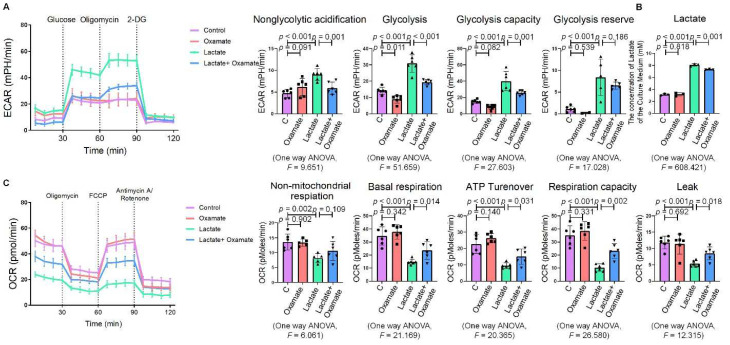
Oxamate attenuated the lactate-induced enhancement of glycolysis in macrophages. (**A**) Glycolysis flux was examined by measuring the ECAR using the Seahorse analyzer. Data are presented as the mean ± SD, *n* = 6 per group. (**B**) The lactate content in the culture medium was detected using a lactate assay kit. Data are presented as the mean ± SD, *n* =3 per group. (**C**) The OCR values were measured using the Seahorse analyzer. Data are presented as the mean ± SD, *n* = 6 per group.

**Figure 7 ijms-23-03013-f007:**
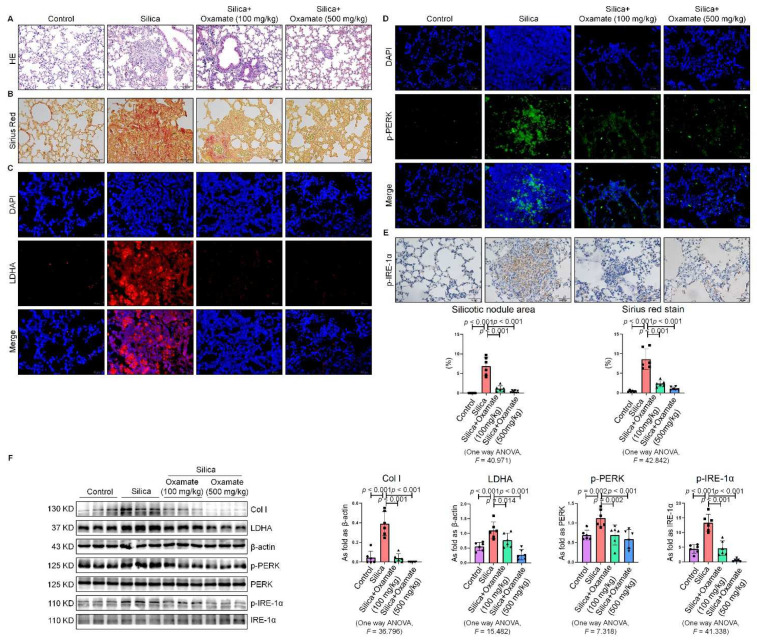
Oxamate reduced glycolysis and ER stress in silicotic mice. (**A**) HE. staining of lung tissue in mice exposed to silica (scale bar = 100 µm). (**B**) Sirius red staining of lung tissue in mice exposed to silica (scale bar = 50 µm). (**C**) Expression of LDHA in mice exposed to silica measured by IF staining (scale bar = 50 µm). (**D**) Expression of p-PERK in silicotic mice measured by IF staining (scale bar) = 50 µm. (**E**) Positive expression of p-IRE-1α in silicotic mice observed by IHC staining (scale bar = 50 µm). (**F**) Expression levels of collagen type I (Col I), LDHA, p-PERK, and p-IRE-1α in mice lungs measured by Western blotting. Data are presented as the mean ± SD, *n* = 6 per group.

## Data Availability

The underlying data of the study can be obtained by contacting the authors if it is reasonable.

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
