# Peer review of "Oxamate Attenuates Glycolysis and ER Stress in Silicotic Mice"

_ijms, 2022, doi:10.3390/ijms23063013_

Round 1
Reviewer 1 Report
The manuscript presents an interesting approach to treat lung silicosis via LDH inhibition. There are some issues in the quantification and presentation of data that need to be addressed before the article can be ready for publication:
In figures 1, 2, 4, and 5, panels A, B, and C, a representative picture is not sufficient to conclude that the different treatment groups have different activation of ER stress. A quantification of the signal in several microscopic fields with statistics should be performed.
In fig. 7A and D, quantifications are needed.
In Figure 1D, the quantification of Western blot signal is not clear. The scale says “As fold as control”, but in that case the control (first lane, “C”) should be 1, and it’s not. Also, it is not very clear if the signal of phosphorylated protein in normalized by total protein. This should be explained better in the figure legend.
In figure 7F, it is not clear which lanes belong to each treatment group. Some horizontal lines over the lanes would be helpful to indicate where one group ends and the other begins.
Author Response
Point 1: In figures 1, 2, 4, and 5, panels A, B, and C, a representative picture is not sufficient to conclude that the different treatment groups have different activation of ER stress. A quantification of the signal in several microscopic fields with statistics should be performed.
Response 1: We used Image-Pro Plus 6.0 (Media Cybernetics, Rockville, MD, USA) to quantitatively analyse the signals of multiple microscopic fields in each group, and supplement the data to the result section in the revised manuscript.
Point 2: In fig. 7A and D, quantifications are needed.
Response 2: The area of silicosis nodules and collagen deposition measured by Sirius red stain were counted by Image-Pro Plus 6.0 software and were homogenized by the total area of lung section. We have supplemented the data to the result section in the revised manuscript.
Point 3: In Figure 1D, the quantification of Western blot signal is not clear. The scale says “As fold as control”, but in that case the control (first lane, “C”) should be 1, and it’s not. Also, it is not very clear if the signal of phosphorylated protein in normalized by total protein. This should be explained better in the figure legend.
Response 3: Thank you for carefully reviewing my article. We have revised the figure legends. This error has been corrected in the revised manuscript.
Point 4: In figure 7F, it is not clear which lanes belong to each treatment group. Some horizontal lines over the lanes would be helpful to indicate where one group ends and the other begins.
Response 4: The related context in result section has been revised.
Reviewer 2 Report
Mao and coworkers presented a paper intitled “Oxamate attenuates glycolysis and ER stress in silicotic mice” where they describe the effect of oxamate using NR8383 macrophage cell line and silicotic mice. The results were well presented and discussed.
Author Response
Point 1: Mao and coworkers presented a paper intitled “Oxamate attenuates glycolysis and ER stress in silicotic mice” where they describe the effect of oxamate using NR8383 macrophage cell line and silicotic mice. The results were well presented and discussed.
Response: Thank you for your advise.